# An integrated genomic regulatory network of virulence-related transcriptional factors in *Pseudomonas aeruginosa*

Hao Huang[1,4], Xiaolong Shao[2,4], Yingpeng Xie[1,4], Tingting Wang[1], Yingchao Zhang[2], Xin Wang ● [1,3] & Xin Deng[1,3]

The virulence of *Pseudomonas aeruginosa*, a Gram-negative opportunistic pathogen, is regulated by many transcriptional factors (TFs) that control the expression of quorum sensing and protein secretion systems. Here, we report a genome-wide, network-based approach to dissect the crosstalk between 20 key virulence-related TFs. Using chromatin immunoprecipitation coupled with high-throughput sequencing (ChIP-seq), as well as RNA-seq, we identify 1200 TF-bound genes and 4775 differentially expressed genes. We experimentally validate 347 of these genes as functional target genes, and describe the regulatory relationships of the 20 TFs with their targets in a network that we call '*Pseudomonas aeruginosa* genomic regulatory network' (PAGnet). Analysis of the network led to the identification of novel functions for two TFs (ExsA and GacA) in quorum sensing and nitrogen metabolism. Furthermore, we present an online platform and R package based on PAGnet to facilitate updating and user-customised analyses.

[1] Department of Biomedical Sciences, City University of Hong Kong, Hong Kong 999077, China. [2] Key Laboratory of Molecular Microbiology and Technology, Ministry of Education, TEDA Institute of Biological Sciences and Biotechnology, Nankai University, 23 Hongda Street, Tianjin 300457, China. [3] Shenzhen Research Institute, City University of Hong Kong, Shenzhen 518057, China. [4]These authors contributed equally: Hao Huang, Xiaolong Shao, Yingpeng Xie. Correspondence and requests for materials should be addressed to X.W. (email: xin.wang@cityu.edu.hk) or to X.D. (email: xindeng@cityu.edu.hk)

*P*seudomonas aeruginosa, an opportunistic human pathogen, tends to cause potentially lethal acute or chronic infections in patients with cystic fibrosis (CF), immuno-compromised individuals and burn victims[1–3]. *P. aeruginosa* harbours multiple key determinants of virulence, including quorum sensing (QS), Type III (T3SS) and Type VI secretion systems (T6SS), flagella and type IV pili. Taken together, these determinants comprise a highly complex and well-ordered virulence system[4–8].

*P. aeruginosa* is a typical Gram-negative bacterium equipped with population-dependent QS systems that synthesise and release small chemical signals to the environment. To date, at least four QS pathways (*las, rhl, pqs* and *iqs*) have been characterised in *P. aeruginosa*, and these were found to be strongly connected and co-regulated[9]. The *las* and *rhl* systems are based on acyl-homoserine lactone (AHL) signals[10,11], whereas the *pqs* system is based on 2-alkyl-4-quinolone (AQ) signals[12]. The regulatory activities of LasR and RhlR are induced by binding to the LasI product *N*-3-oxo-dodecanoyl-homoserine lactone (3OC12-HSL) and the RhlI product *N*-butanoyl-homoserine lactone (C4-HSL), respectively. The IQS signal molecule is synthesised by the *ambBCDE* operon[13]. The DNA binding affinity of IQS is induced through interactions with IqsR. Subsequently, IQS regulates the production of PQS, C4-HSL, rhamnolipids, elastase and pyocyanin[13]. These accumulated signals finely tune the transcription of hundreds of target genes, including those encoding virulence factors[14]. The QS system in *P. aeruginosa* controls a large group of genes involved in biofilm formation, motility, cytotoxicity and antibiotic resistance[6,15]. Several key regulators (such as PhoB, GbdR, PchR, SphR, ExsA, SoxR and BfmR) were found to strictly regulate QS and responses to environmental changes. Our recent studies elucidated the regulatory mechanisms of five regulators associated with QS, including VqsR[16], VqsM[17], AlgR[18], CdpR[19] and RsaL[20]. The findings suggest that these regulators form a complex regulatory network.

The virulence of *P. aeruginosa* can also be controlled by the T3SS and T6SS. In this bacterium, the T3SS is self-equipped with a needle-shaped supramolecular apparatus used to transport T3SS effectors to host cells[21]. ExsA, an AraC family regulator of transcription[22], controls the expression of most T3SS genes. The expression of ExsA is self-regulated via binding to an adenine-rich region upstream of the putative -35 RNA polymerase binding site[23]. The T3SS is positively regulated by PsrA[24], HigB[25], Vfr[26] and DeaD[27], but negatively regulated by MexT[28], AlgZR, GacAS/LadS/RetS[7,29–33] and MgtE[34]. The *P. aeruginosa* T6SS uses a bacteriophage-like structure with a sharp spike and an outer contractile sheath to translocate proteins into neighbouring prokaryotic or eukaryotic cells[35]. *P. aeruginosa* harbours 3 different T6SS loci: H1-T6SS, H2-T6SS and H3-T6SS[36–38]. Similar to the T3SS, the T6SS is tightly regulated by the GacAS/Rsm pathway. Once activated by GacAS, RsmY/Z positively regulates both H1-T6SS and H3-T6SS by inhibiting the binding activity of RsmA or RsmN to *fha1* and *tssA1*[30,39]. TseF, which is secreted by H3-T6SS, is incorporated into outer membrane vesicles (OMVs) with assistance from PQS. Subsequently, TseF facilitates the import of the PQS–$Fe^{3+}$ complex into cells[40].

Transcription factors (TFs) are DNA-binding proteins that control downstream gene expression by promoting or blocking the recruitment of RNA polymerase to specific genes[41]. These TFs bind to gene promoters and closely coordinate the initiation of transcription in response to environmental conditions[41]. A recent study of 10 sigma factors (AlgU, FliA, RpoH, RpoN, RpoS, PvdS, FpvI, FecI, SigX and FecI2) in *P. aeruginosa* revealed that these factors formed a network with an exquisite modular architecture[42]. Another study summarised the known targets of regulators related to *P. aeruginosa* virulence[3]. To date, several

virulence-related TFs of *P. aeruginosa*, including VqsM[17], AlgR[18], AmrZ[43], CdpR[19], RsaL[20], BfmR[44], VqsM[16], MvfR[45] and LasR[46], have been studied individually. However, little is known or verified about the crosstalk between strictly virulence-related TFs in this bacterial species.

In this study, therefore, we performed a genome-wide characterisation of the regulons of 20 virulence-related TFs involved in virulence regulatory processes such as QS, T3SS and T6SS, based on an integrative analysis of transcriptome profiling (RNA-Seq) and chromatin immunoprecipitation coupled with high-throughput sequencing (ChIP-seq) profiles. Using an integrative network approach, we mapped a *Pseudomonas aeruginosa* Genomic regulatory network, PAGnet, which encodes the regulatory relationships of these 20 TFs with their functional targets. Subsequently, crosstalk between the TFs in the PAGnet, defined as the co-regulation of genes by at least 2 TFs, was experimentally verified using electrophoretic mobility-shift assay (EMSA) and real-time quantitative PCR (RT-qPCR) analyses. This network not only revealed the master regulators of individual pathways, but also genes that participate in crosstalk and are thus involved in multiple virulence and metabolic pathways. Finally, we developed an online platform and a downloadable R package to enable network visualisation and analysis. We expect that these tools will significantly facilitate future studies of the global regulation of virulence in *P. aeruginosa*.

## Results

**Genome-wide DNA binding patterns of virulence-related TFs in *P. aeruginosa*.** Although previous studies have demonstrated the individual contributions of 20 key TFs to the virulence of *P. aeruginosa*, including AlgR, AmrZ, BfmR, CdpR, GacA, LasR, MexT, MvfR, QscR, RhlR, RsaL, SoxR, VqsM, VqsR, PhoB, GbdR, PchR, SphR and FleQ (their known functions are summarised in Supplementary Table 1), the potential for crosstalk between these factors remains largely unknown. In the present study, we aimed to reveal the genome-wide landscape of crosstalk between these virulence-related TFs and eventually map an integrated regulatory network of the *P. aeruginosa* genome.

To systematically investigate the virulence regulatory pathways of these 20 TFs in *P. aeruginosa*, we first performed genome-wide ChIP-seq analyses to identify the direct targets of the TFs on a genomic scale. As the target genes of AmrZ, LasR, VqsR and RsaL were previously characterised[16,20,43,46], this study newly characterised the genome-wide protein-DNA binding patterns of the remaining 15 TFs, including VqsM[17], AlgR[18], CdpR[19], MvfR, RhlR, GacA, ExsA, MexT, QscR, SoxR, PhoB, GbdR, PchR, SphR and FleQ. Details of the ChIP-seq experiment are discussed in the Methods section and Supplementary Table 2. The VSV-G-tagged RsaL and MvfR constructs functionally complemented their corresponding deletion strains in a pyocyanin production assay[20,47] (Supplementary Fig. 1a). A Congo Red assay of colony morphology also revealed that FLAG-tagged FleQ was functionally complementary in a ∆*fleQ* strain[48] (Supplementary Fig. 1b). For each TF, the raw ChIP-seq reads were mapped to the *P. aeruginosa* genome using Bowtie 1.2.2[49], and enriched loci harbouring TF-binding peaks were identified using MACS 2.0 ($P < 1 \times 10^{-5}$)[50]. Subsequently, the enriched loci for each TF were annotated using the R package ChIPpeakAnno[51], and these data were used to globally characterise TF-binding peaks on the *P. aeruginosa* genome (Fig. 1a). GacA yielded the highest number (1125) of binding peaks. Of these peaks, 31.6% were detected in the intergenic regions upstream of genes (upstream) and 4.4% in the intergenic regions downstream of genes (downstream); additionally, 2.3% overlapped with the translation start sites of genes (overlapStart), 2.1% overlapped with the ends of genes

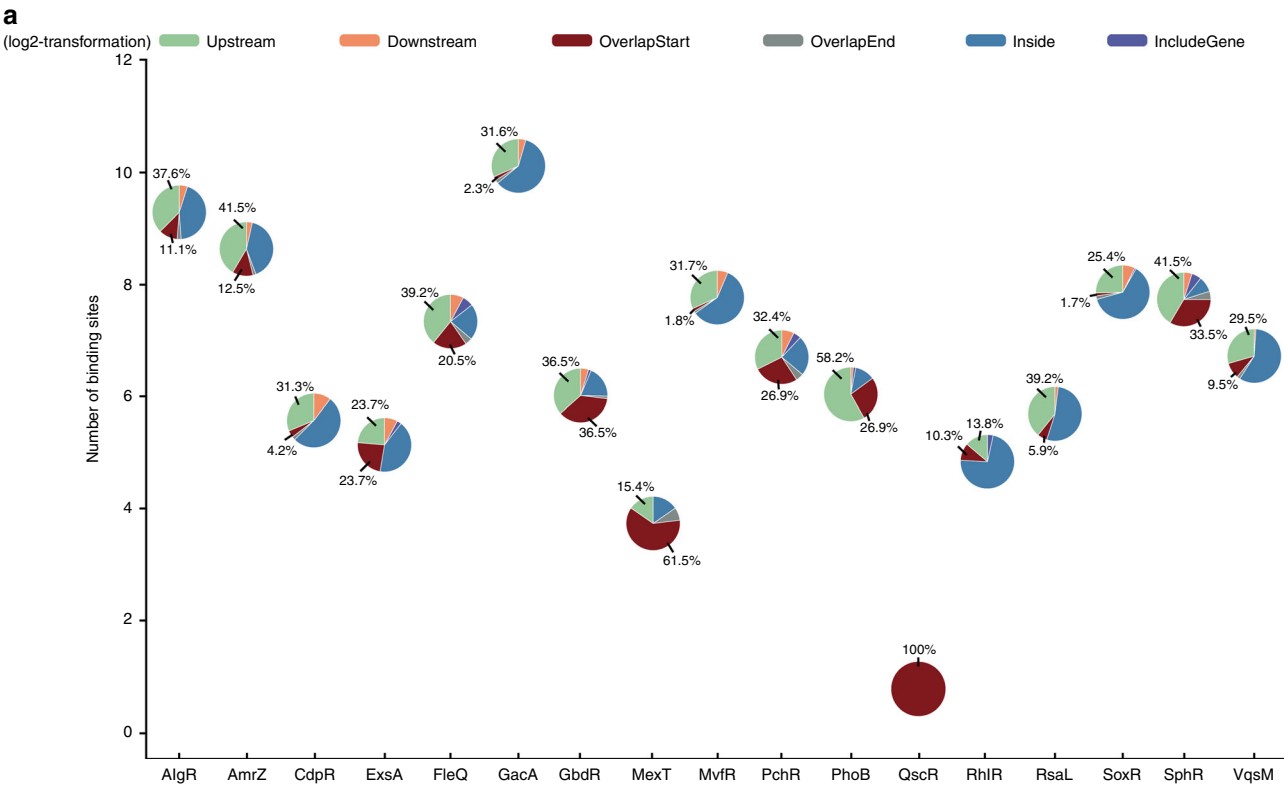

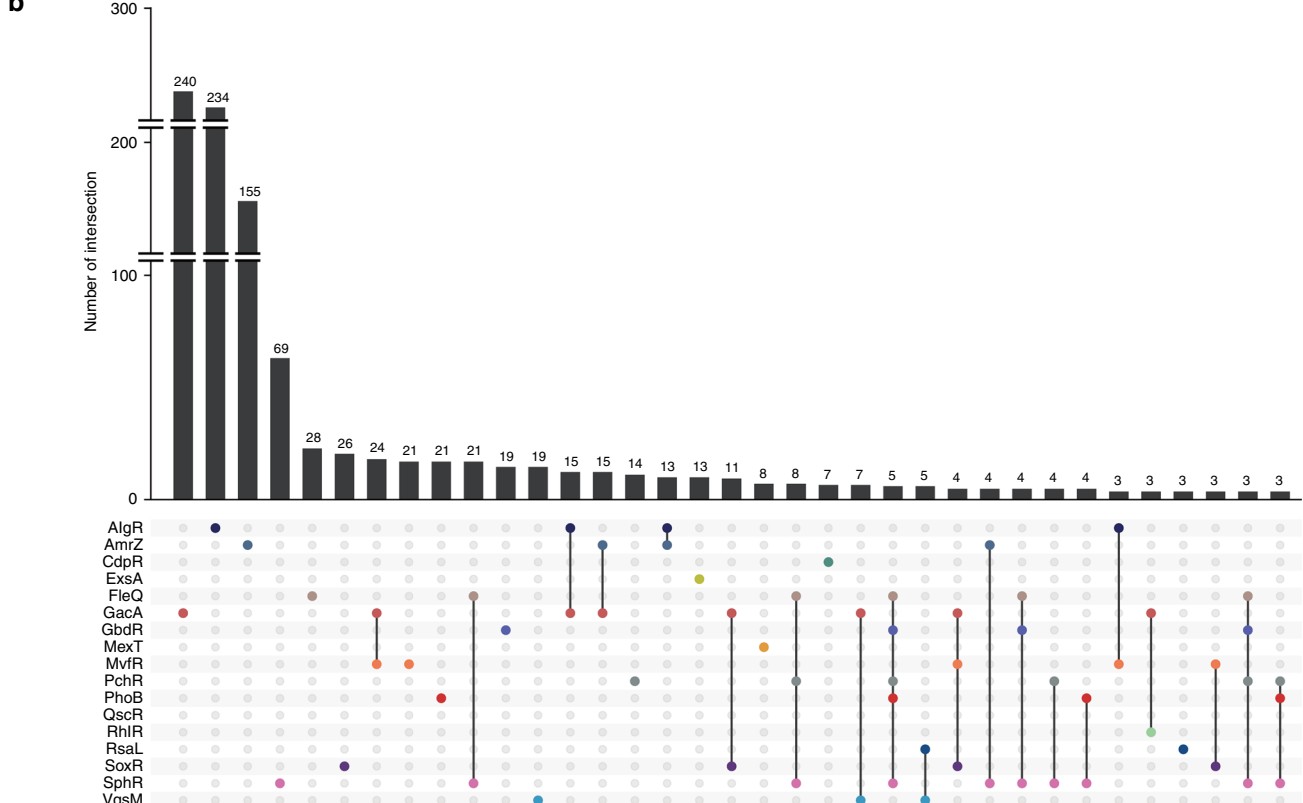

**Fig. 1** TF-binding peak annotations and the overlap of peak target genes. **a** The position annotation of different TF-binding peaks in *P. aeruginosa* genome using pie charts, and horizontal axis (log2-transformation) represents the number of binding peaks of TFs. **b** The overlap of different TFs target genes (TF-binding peaks located on promoters of genes), the histogram represents the number of genes in individual/overlapped set

(overlapEnd) and 59.6% resided within coding regions (inside) (Fig. 1a). AlgR yielded fewer binding peaks, and most were classified as inside (43.9%). QscR only yielded one binding peak that overlapped with the translation start site of PA1898. Altogether, 17 TFs (including AmrZ) yielded a total of 3479 binding peaks and exhibited different binding preferences throughout the genome, which suggested different regulatory functions.

Here, we mainly focused on 1200 annotated genes that contained binding peaks in their promoter regions (i.e., intergenic region upstream of a gene) and were directly regulated by 1 or more TFs (Supplementary Data 1). Most of these genes (935/1200) were targeted by only 1 TF, while the remaining 265 genes were co-targeted by multiple TFs (co-targets). These findings indicate the complicated binding patterns and potential functional crosstalk among these virulence-related TFs (Fig. 1b and Supplementary Fig. 2). In particular, GacA and AlgR respectively bound to 240 and 234 unique genes (Fig. 1b). GacA and MvfR bound jointly to 24 genes, suggesting that these TFs co-regulate many pathways. Five genes were jointly bound by 5 TFs, including FleQ, GbdR, PchR, PhoB and SphR (Fig. 1b), and the *phzA1* promoter was co-bound by 6 TFs (AlgR, FleQ, GbdR, PchR, PhoB and SphR). Five TFs co-bound to the promoter of a group of genes that included *algD* (involved in alginate biosynthesis and biofilm formation), *fimV* (encoding the motility protein FimV), *phdA, rhlR, rocA2* (involved in antibiotic resistance), *acpP* (encoding an acyl-carrier protein) and 6 hypothetical protein encoding genes (PA4139, PA1333, PA3835, PA4087, PA4340 and PA4676). An additional 20 promoters were co-bound by 4 TFs. In other words, these TFs appear to be linked intricately. All genes bound by multiple TFs are listed in Supplementary Data 1.

Figure 2a presents a genome-wide overview of the binding peaks of every TF throughout the *P. aeruginosa* genome. The binding loci of these TFs were distributed throughout the genome, and specific patterns were observed for individual TFs (Fig. 2a). The consensus motifs of 16 TFs were also identified based on the binding sequences from ChIP-seq using MEME[52] (Fig. 2b). QscR had only 1 peak and was deemed insufficient for the motif analysis. The individual binding motifs of VqsM, AlgR, CdpR and RsaL were identified in our previous studies[17–20,45,53]. Although the binding motifs of ExsA, SoxR, MexT and MvfR[45] were previously determined using a multiple sequence alignment[54–57], our ChIP-seq-based analyses further explored and verified these motifs throughout the genome. Accordingly, we identified 3 motifs that contained repetitive sequences, including a 14-bp GacA-binding motif (CGNCCAGGNCCAGG), a 16-bp PhoB-binding motif (ATGACNNNTNNATGAC) and a 12-bp PchR-binding motif (C/TGGC/TGCTG/TGCGG) (Fig. 2b).

**Transcriptome characterization of virulence-related TFs in *P. aeruginosa*.** As we observed profound crosstalk among the tested TFs, we next performed an integrated characterisation of their regulons. The transcriptomes of 10 of the 20 TFs (AlgR[58], AmrZ[59], BfmR[44], VqsR[60], ExsA[61], GacA[62], MexT[57], RsaL[63], SoxR[56] and VqsM[64]) under different conditions and growth phases were previously characterised through RNA-seq or microarray analyses. In this study, we first standardised the experimental conditions for all transcriptomes as an optical density at 600 nm ($OD_{600}$) of 0.6 and culture in Luria-Bertani (LB) medium. We repeated the RNA-seq experiment with 16 PAO1 deletion strains ($\Delta phoB$, $\Delta gbdR$, $\Delta pchR$, $\Delta sphR$, $\Delta fleQ$, $\Delta cdpR$, $\Delta gacA$, $\Delta rsaL$, $\Delta vqsR$, $\Delta exsA$, $\Delta lasR$, $\Delta mvfR$, $\Delta rhlR$, $\Delta qscR$, $\Delta vqsM$ and $\Delta algR$) under the same conditions (LB, $OD_{600} = 0.6$). Detailed information about RNA-seq experiment is provided in

the Methods section and Supplementary Table 2. Because our previous study showed that BfmR only regulates *rhlQS* in M8-glutamate minimal medium but not in LB medium[65], we did not obtain additional RNA-seq data under our set conditions (LB, $OD_{600} = 0.6$). Similarly, we did not subject SoxR to a repeat RNA-seq analysis because this TF mainly functions under oxidative stress[56].

The integrated analysis of all RNA-seq and microarray data revealed that 4775 of 5704 genes (83.7%) in the *P. aeruginosa* genome were affected by these 20 TFs (Fig. 3). The overlapping distribution patterns of all target genes controlled by these 20 TFs and their individual patterns are shown in Fig. 3a. These patterns revealed an intricate network of complicated co-regulations, where in most regulatory genes were co-regulated by multiple TFs. A total of 1297 genes were regulated by a single TF, whereas 3478 genes were regulated by multiple TFs (Supplementary Data 2). For example, LasR shared 301, 98, 25, 14, 4 and 3 co-targets with BfmR, AmrZ, GacA, RsaL, VqsM and SphR, respectively (Fig. 3b and Supplementary Fig. 3). Among the regulated genes, the gene *nosZ* (encoding a nitrous-oxide reductase precursor) was co-regulated simultaneously by 16 regulators (AlgR, AmrZ, BfmR, CdpR, FleQ, GacA, GbdR, MvfR, PchR, PhoB, QscR, RhlR, RsaL, SphR, VqsM and VqsR). The expression of *lasB* (encoding elastase) was co-regulated by 14 TFs (AlgR, AmrZ, BfmR, ExsA, FleQ, GacA, GbdR, LasR, MexT, PchR, PhoB, QscR, RsaL and SphR), while *narK1* was co-regulated by 13 regulators (AlgR, AmrZ, BfmR, ExsA, FleQ, GacA, GbdR, PchR, PhoB, RhlR, RsaL, SphR and VqsR). In addition, more than 10 operons (such as *phzA1-G1, phzA2-G2, hcnA-C, pqsA-E, mexGHI-OmpD, mexEF-oprN, nirS-nirN, arcCBAD, antABC* and *narK1-narI*) were jointly regulated by multiple TFs (Supplementary Data 2). For instance, the *phzA1-G1* operon was found to be co-regulated by AmrZ, BfmR, ExsA, GacA, GbdR, LasR, MexT, MvfR, PchR, PhoB, RhlR and RsaL. The *pqsA-E* operon was co-regulated by AmrZ, ExsA, GacA, GbdR, LasR, MexT, MvfR, PchR, PhoB and RsaL, which are the key determinants of virulence in *P. aeruginosa*. Taken together, our integrated transcriptional profiling revealed a complex and multifaceted regulatory network involved in virulence and metabolism, suggesting the presence of intimate connections between virulence-associated TFs in *P. aeruginosa*.

**Mapping a functionally integrated regulatory network.** In order to more comprehensively investigate direct functional targets of the 20 virulence-related TFs, all ChIP-seq and RNA-seq data obtained in this and previous studies were integrated to identify their combined regulome. Here, we defined a gene as a functional target of a particular TF if it fulfilled the following criteria: (1) its upstream region was bound by TFs and (2) its expression level was significantly affected by the deletion of TFs. The virulence-related TF regulome revealed 347 functional target genes, including 42 genes as functional targets controlled by multiple TFs (Supplementary Fig. 4 and Supplementary Data 3). Notably, we found that LasR directly co-regulated the expression of 14 genes (such as *lasI, lasB, hcnA-C, cdpR, pqsA-E, rhlR, pslA* and *amrZ*) together with 11 other TFs. Thirteen genes were co-regulated by AmrZ or PhoB with other TFs. Particularly, *phzA1-G1* was directly co-regulated by 7 TFs, including CdpR, GbdR, PchR, PhoB, QscR, RhlR and RsaL. Three QS genes (*lasI, hcnA-C* and *cdpR*) were also co-regulated by 4 TFs (Supplementary Data 3).

Next, we mapped an integrated virulence regulatory network, 'PAGnet' (Fig. 4), to visualise the interactions between the 20 regulatory TFs associated with virulence and the inferred functional targets. The PAGnet included 20 TFs and 347

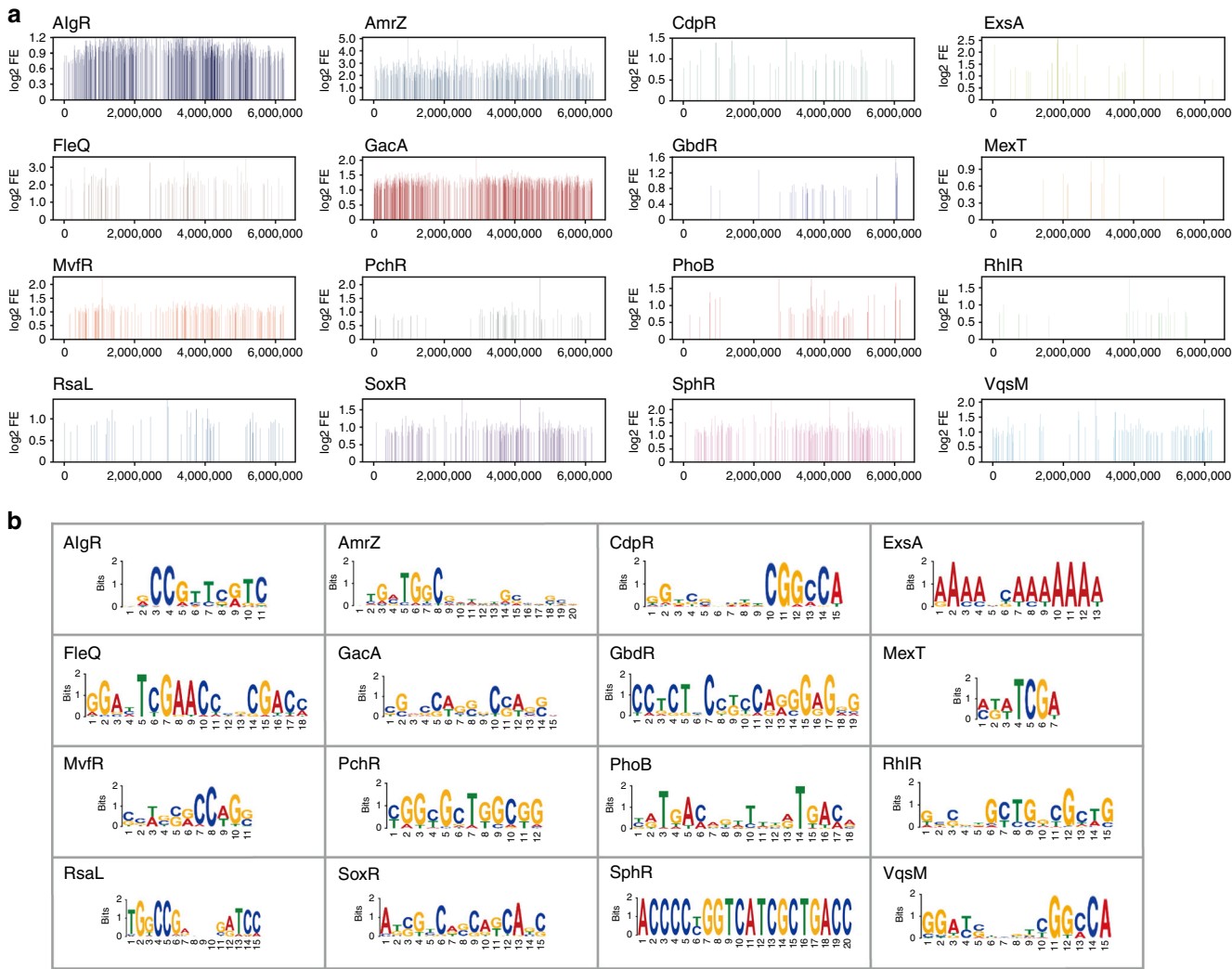

**Fig. 2** Genomic location and binding motif analysis of 20 TFs. **a** The coverage of 16 transcription factors peak regions over *Pseudomonas aeruginosa* chromosome. Each line shows the location and signal enrichment value (log2 Fold Enrichment) of peaks of a transcription factor peak in chromosome. **b** The binding motifs of transcriptional factors were elucidated using MEME. All peaks were used to define the binding motif. The height of each letter presents the relative frequency of each base at different positions in the consensus sequence

functional target genes interconnected by 409 functional interactions (Supplementary Data 3). In PAGnet, TF-mediated regulation of a functional target gene can occur in either a direct (TF1-target) or an indirect (TF1-TF2-target) manner. Genes co-regulated by multiple TFs were involved in various biological pathways, including QS (*rhlR, rhlA, rhlB, cdpR, pqsH, pqsA, rsaL, lasI* and *lasB*), pyocyanine biosynthesis (*phzA1-G1* and *phzA2-G2*) T6SS (*amrZ* and *hsiA2*), metabolism (*arnB, glyA3, pchG, roeA* and *asd*), signal transduction (*phnC*), biofilm formation (*pslA*) and unknown functions (PA4139, PA1159, PA173, PA2228, PA3691 and PA4087) (Supplementary Data 3).

To identify the potential functions or pathways associated with the regulon of each TF in the PAGnet, we performed functional annotation using hypergeometric tests (BH-adjusted $P < 0.05$) based on PseudoCAP[66] and gene sets from Gene Ontology (GO) and Kyoto Encyclopaedia of Genes and Genomes (KEGG) databases (Fig. 5). The functional categories associated with the regulons of all 20 TFs are listed in Supplementary Data 4. The functional categories associated with the regulons of all 20 TFs (Supplementary Data 4.) showed divergent functions. For example, the regulons of CdpR, GbdR, MvfR, PchR, PhoB, QscR, PhlR and RsaL regulated phenazine biosynthesis, while AlgR,

ExsA, GacA, LasR, MvfR, RhlR and VqsM were involved in biofilm formation.

Furthermore, a master regulator analysis (MRA)[67] based on PAGnet and virulence-related gene signatures was performed to identify the master regulators of the QS, T3SS and T6SS pathways. Accordingly, RsaL, QscR, RhlR, CdpR, MvfR, PchR, PhoB and LasR were identified as master regulators of QS in our network (Hypergeometric test, BH-adjusted $P < 0.05$) (Supplementary Fig. 5). ExsA was identified as the master regulator of T3SS (Hypergeometric test, BH-adjusted $P < 0.05$) (Supplementary Fig. 6).

**Experimental verification of functional targets of TFs**. To verify the functional targets of TFs in the PAGnet, we subjected a group of randomly chosen genes to EMSA and RT-qPCR. Here, PhoB indeed bound to the promoters of PA4139, *xcpR* (encoding general secretion pathway protein E), *nosR* (encoding nitrous-oxide reductase expression regulator), PA0123, PA4108, *phoB* (positive control), *PA1736, narK1* and *rhlR* (Fig. 6a–e and Supplementary Fig. 7a–d). FleQ bound to the promoter of PA4139, PA3520, PA3662, *arnB* and

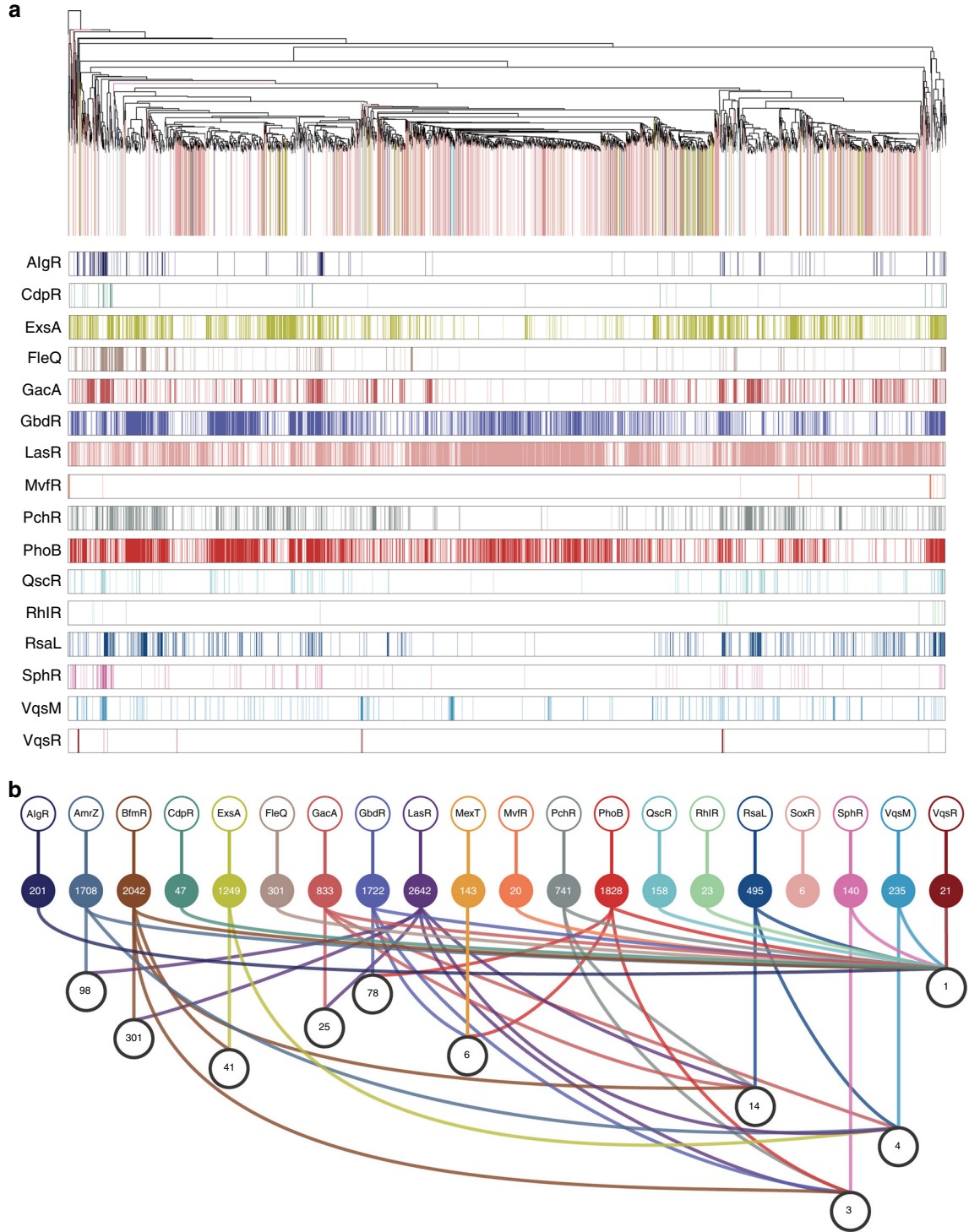

**Fig. 3** Clustering and co-expression patterns of virulence-related TFs. **a** Hierarchical clustering based on differentially expressed genes in absence of TFs using Pearson correlation coefficients. 4775 genes were differentially regulated by 16 transcription factors obtained from our RNA-seq. Genes regulated by more than one TFs are coloured in white. **b** Co-expression patterns of differentially regulated genes dependent upon 20 TFs, white balls lined between different TFs balls represent the number of genes that are co-regulated by the TFs combinations

*pelA* (positive control) (Fig. 6f–i and Supplementary Fig. 7e). AlgR also bound efficiently to the promoter of *phdA* (encoding prevent-host-death protein A) and *mucR* (positive control) (Fig. 6j and S7f). Both PhoB and FleQ bound to the promoter of PA4139, indicating a co-regulatory process (Fig. 6a, f). LasR bound to the promoter of *xcpR*[46]. In addition, SoxR interacted physically with the promoters of PA2274 (positive

control), *mexG*, PA3718, *mqoB*, *oruR*, PA4330 and *mexS* (Supplementary Fig. 7g–m). AmrZ bound to the promoters of *gcbA*[43] (positive control), *flp, flgG, dctP* and *fleQ* (Supplementary Fig. 7n–r). The negative controls, PhoB, FleQ, AlgR, SoxR and AmrZ, were assessed at equivalent protein concentrations. No binding bands were observed (Supplementary Fig. 7s–w).

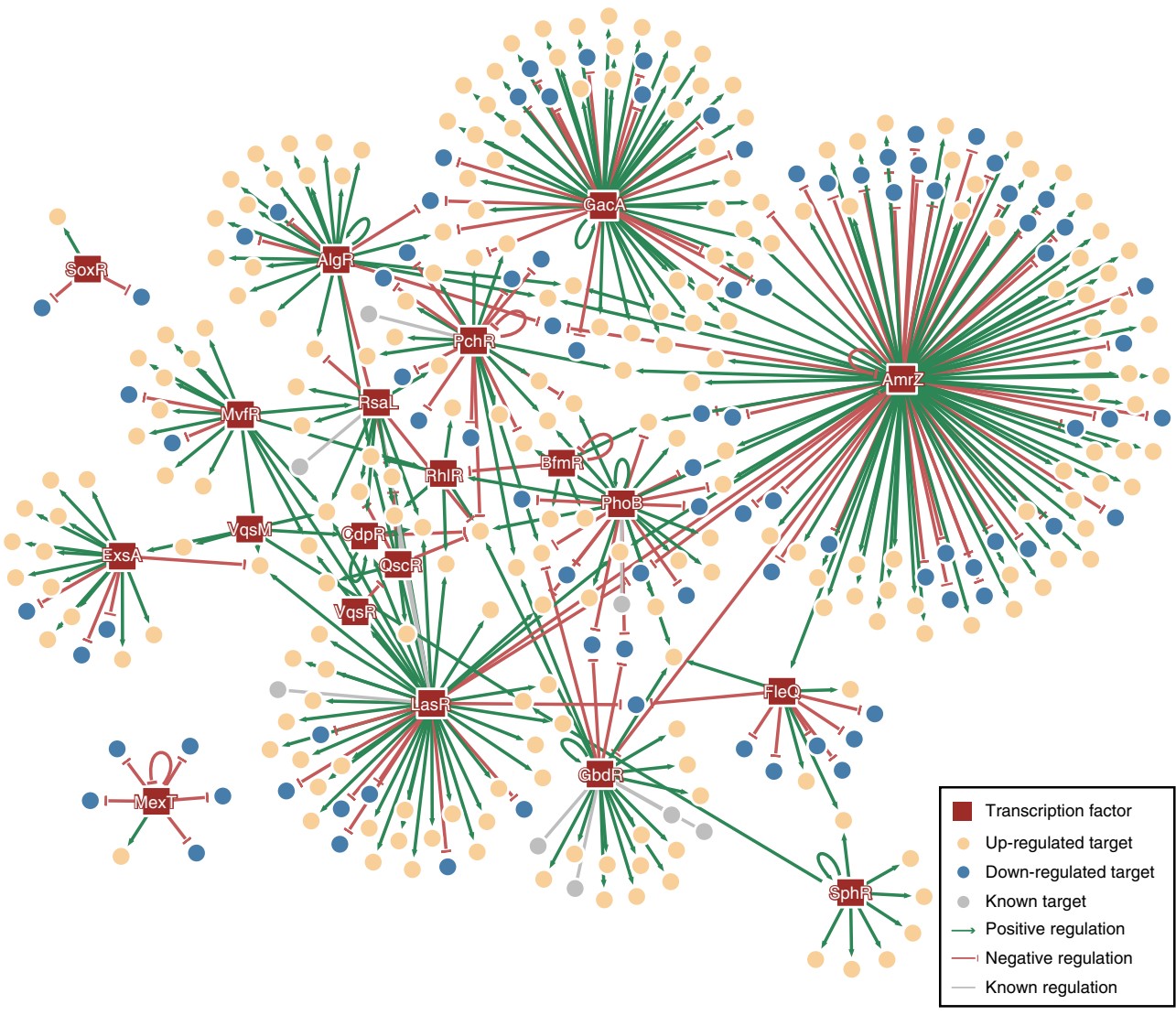

**Fig. 4** Visualization of the PAGnet. PAGnet is established by integrated twenty *Pseudomonas aeruginosa* transcriptional factors regulons. The rectangle represents 20 genomic transcription factors, and the circle represents the functional targets of different transcription factors. Among these targets, the orange targets are positively regulated by the transcription factor, the blue ones are negatively regulated by the transcription factors, and the grey ones represent regulatory relationship is unknown. The target can be co-regulated by multiple transcription factors, and some TFs have self-regulation

RT-qPCR for validation revealed that the PAGnet yielded a good performance (accuracy: 85.7%, precision: 84.4% and recall: 100%) (Fig. 6, Supplementary Figs. 7 and 8). Taken together, our experiment verified the PAGnet as a reliable database with which to explore the direct targets of tested TFs in *P. aeruginosa*.

**The T3SS master regulator ExsA regulated non-T3SS pathways**. ExsA, a master regulator of T3SS in *P. aeruginosa*, contributes to acute virulence phenotypes[45]. In our ChIP-seq analysis, ExsA bound directly to 17 promoter regions. Sixteen of these loci were regulated by ExsA, and 5 were identified as T3SS-related genes (encoding needle complex proteins, the translocation apparatus, regulatory proteins, the effector proteins and the chaperones) (Supplementary Fig. 6). In 3 newly identified T3SS targets, PopN and Pcr1 formed a complex that represses the T3SS[68]. Here, ExsA bound directly to the *popN* promoter via the conserved ExsA-binding motif and positively regulated the transcription of *popN* and *exsC* (positive control) (Fig. 6k and S8a). ExsA also positively regulated PA3842 and PA3840 by binding to their promoters (Fig. 6l and Supplementary Fig. 8b).

In addition, ExsA bound 4 promoters of non-T3SS genes. Specifically, it negatively regulated *fabG* (encoding 3-oxoacyl-[acyl-carrier-protein] reductase), which is involved in lipid synthesis (Supplementary Fig. 8c) and is considered as a potential drug target[69]. ExsA also negatively regulated *ccoN2* (encoding cbb3-type cytochrome oxidase), which plays a role bacterial colonisation (Fig. 6m), and the small RNA PhrS (Supplementary Fig. 8d), which is associated with oxygen availability and quorum sensing[70]. In comparison, the same concentration of ExsA did not bind to the *hcpA* promoter (negative control) (Supplementary Fig. 8e). ExsA also bound to the promoter of *fliK* (encoding the flagellar export switching machinery component protein FliK), *dbpA* (encoding RNA helicase DbpA), *rho* (encoding Rho-dependent termination protein) and *wbpH* (encoding glycosyltransferase) and *wbpA* (encoding UDP-N-acetyl-d-glucosamine 6-dehydrogenase). Taken together, these results strongly suggest that ExsA plays important regulatory roles in pathways other than T3SS.

**Multiple regulatory interactions of GacA in the PAGnet**. Our ChIP-seq analyses revealed that GacA bound to its own promoter

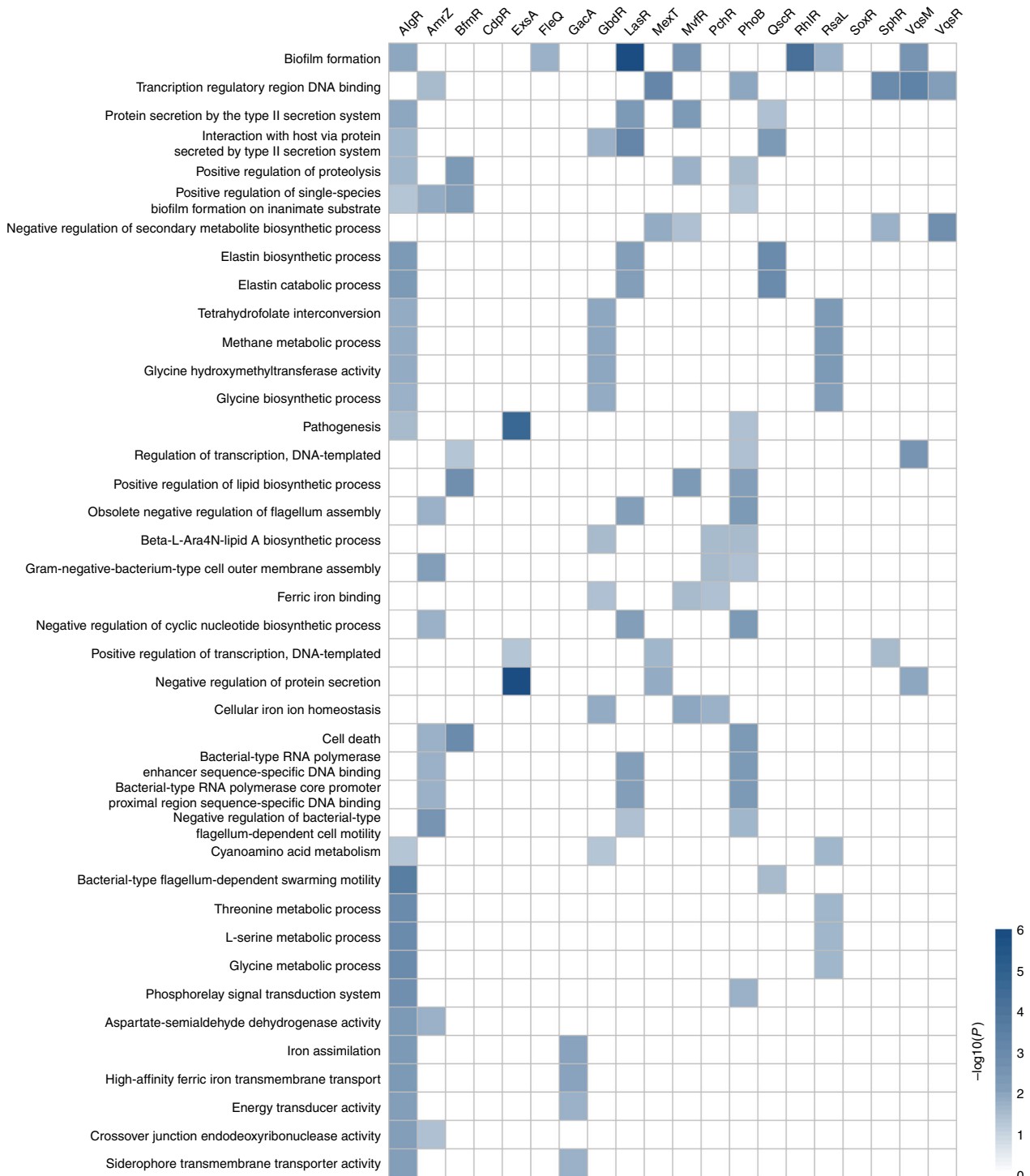

**Fig. 5** Functional characterization of the PAGnet. The PseudoCAP annotation was used to categorize the targets of 20 transcription factors in network. The colour shade of different block indicates the significance of each TF on every functional category (-log10 (*P*))

to self-regulate its expression. We further confirmed high-affinity binding of GacA directly to its own promoter via EMSA (Supplementary Fig. 8f). More importantly, GacA regulated the *pqs*-QS system by binding directly to the *pqsH-cdpR* intergenic region and the *pelB* promoter to control biofilm formation (Fig. 6n and Supplementary Fig. 8g). GacA also bound directly to *glnD* (encoding protein-PII uridylyltransferase) and *nasA* (encoding nitrate transporter) promoters to regulate adaptation to nitrogen stress (Fig. 6o, p). In addition, GacA bound to the promoters of *cbpD* (encoding chitin-binding protein CbpD precursor), *magD*

(encoding endopeptidase inhibitor), *mifS* (involved in α-Keto-glutamate utilisation) and *napF* (encoding ferredoxin protein) (Fig. 6q, r and Supplementary Fig. 8h, i). GacA at the same protein concentration did not interact with the *mexG* promoter (negative control) (Supplementary Fig. 8j).

The PAGnet also revealed 6 genes that were directly co-regulated by GacA and other TFs (Supplementary Data 3). For example, PA1107, PA4224 and PA4488 were co-regulated by GacA and AmrZ. *PA3637* and *PA5531* were co-regulated by GacA and AlgR. PA4221 was co-regulated by GacA and PchR. In

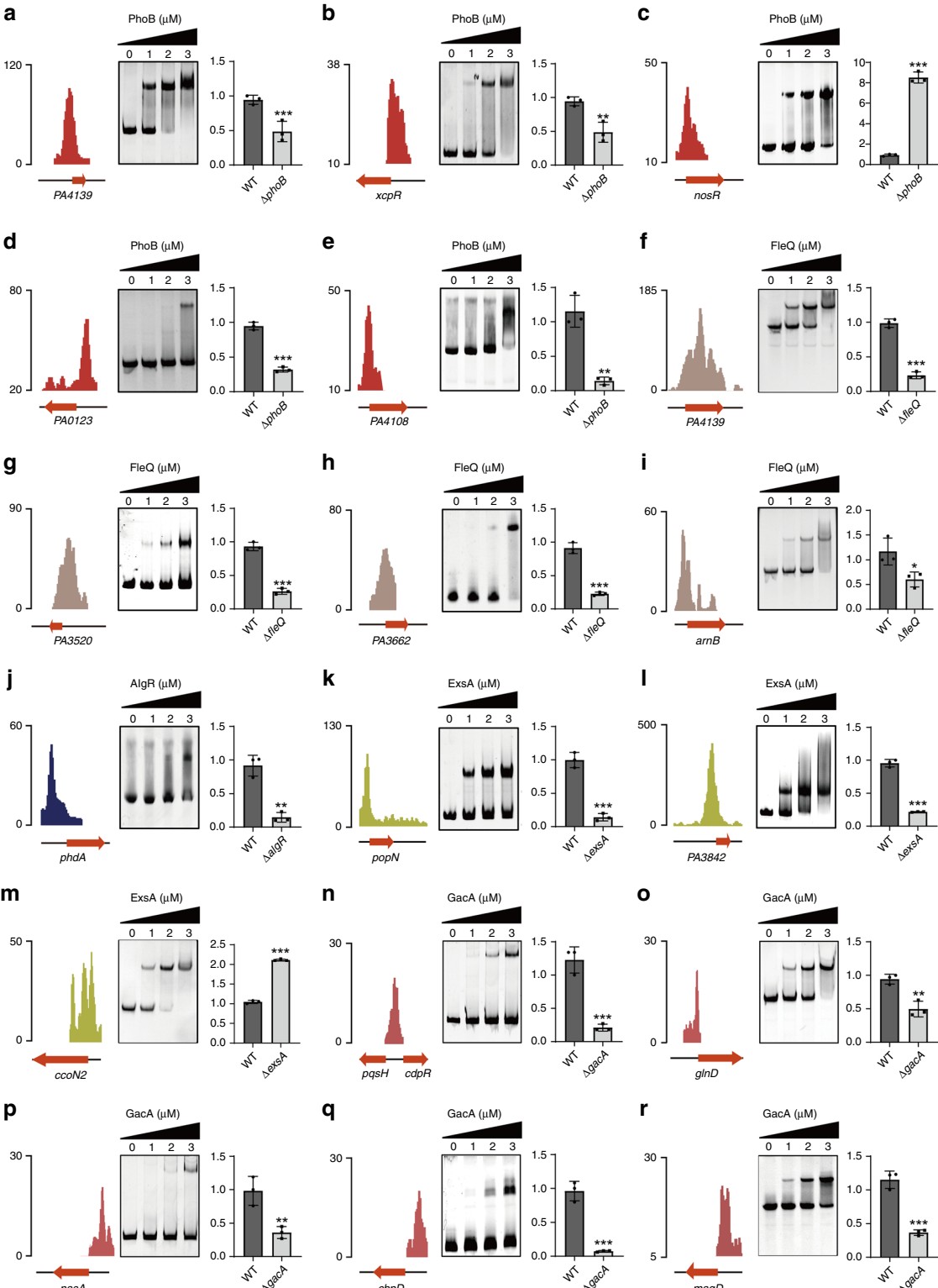

**Fig. 6** Verification of the functional targets of TFs by EMSA and qRT-PCR. The original sequence peaks show the TFs binding regions, and TFs binding regions (PCR-amplified from *P. aeruginosa* PAO1 genome) identified by ChIP-seq were mixed with an increasing amount of purified TFs protein for the EMSA assay. The expression of target genes was examined by RT-qPCR at the same time. PhoB directly regulated the transcription of PA4139 (**a**), *xcpR* (**b**), *nosR* (**c**), PA0123 (**d**) and PA4108 (**e**). FleQ directly regulated the expression of PA4139 (**f**), PA3520 (**g**), PA3662 (**h**) and *arnB* (**i**). AlgR directly regulated expression of *phdA* (**j**). ExsA directly regulated the expression of *popN* (**k**), PA3842 (**l**) and *ccoN2* (**m**). GacA directly regulated the expression of *cdpR* (**n**), *glnD* (**o**), *nasA* (**p**), *cbpD* (**q**) and *magD* (**r**). All experiments were repeated at least three times. Two-tailed Student's *t*-tests were used to examine the mean differences between the data groups. *$P < 0.05$, **$P < 0.01$ and ***$P < 0.001$. Error bars show standard deviations

summary, GacA regulated multiple pathways and interacted with other TFs in the PAGnet.

**PAGnet online platform**. A web application implementing PAGnet (Supplementary Fig. 9a) was deployed as a freely accessible online platform at http://pagnetwork.org/. This platform provides network visualisation, subnetwork filtering and downloading services to the user (Supplementary Fig. 9b). More specifically, the platform uses an optimised, dynamic layout based on *visNetwork* package to facilitate the visualisation and exploration of a virulence regulatory network. This layout allows the user to filter the full network by selecting 1 or more transcription factor(s) to obtain a subnetwork of interest. A brief summary of the subnetwork, with information about the TFs and their target genes, is also provided.

In addition to these basic functions, the PAGnet online platform also enables analysis of master regulators for the identification of key TFs that mediate a biological process or pathway (Supplementary Fig. 9c). First, the user can select the default PAGnet or upload their own regulatory network in a predefined format. Second, the user will specify a gene signature associated with a biological function or pathway of interest, either by selecting a gene set from public databases or uploading a user-customised gene list. The current version of the platform provides gene sets from gene ontology (GO) and KEGG databases obtained from the *Pseudomonas* Genome DB. Having completed master regulator analysis, a table will be returned with information about the corresponding gene ID, gene name, number of target genes, total number of hits (all signature genes in the network), observed hits (signature genes in the regulon of the TF) and a *p*-value calculated based on a hypergeometric test for each TF. This table will be sorted according to the statistical significance indicated by the p-values, allowing the prioritisation of the top significant TFs as master regulators.

To provide more flexibility, we also made the PAGnet available as an R package, 'PAGnet', on GitHub (https://github.com/CityUHK-CompBio/PAGnet). This package can be freely downloaded, locally installed and run by the user on his/her personal computer. The R package includes all functionalities available on the 'PAGnet' website and is powered by a local Shiny-based graphical user interface (GUI) within R. Moreover, the master regulator analysis can be performed using the function 'pagnet.mra' in the R console without running the GUI. More detailed instructions about installation and a step-by-step guide for use are provided in the GitHub-hosted vignette.

## Discussion

Although the virulence-related TFs in the *P. aeruginosa* QS system, T3SS and T6SS have been widely characterised in recent decades, the integrated network and the interactions among these TFs remain elusive. Previously, we and our collaborators elucidated several QS-related TFs, including VqsR[16], VqsM[17], AlgR[18], CdpR[19] and RsaL[20]. In the present study, we established a PAGnet and defined the primary regulons of 20 key virulence-related TF to reveal functional crosstalk and the master regulators of QS and the T3SS. Transcriptional profiling of these 20 virulence-related TFs revealed 3463 intersections, suggesting that these TFs co-regulate their downstream genes in a complex and delicate manner.

Previously, a fine regulatory network of 10 sigma factors (AlgU, FliA, RpoH, RpoN, RpoS, PvdS, FpvI, FecI, SigX and FecI2) revealed the crosstalk that orchestrates complex cellular processes in *P. aeruginosa*[42]. Additionally, a global overview of virulence gene regulation in *P. aeruginosa* has been summarised in a review[3]. Compared with those 2 studies, the present study revealed the following new information. (1) Although the study by Schulz et al.[42] focused on

sigma factors in PA14, the present study focused strictly on TFs in PAO1. (2) Most importantly, we reported a large set of new data which was not included in the previous 2 studies. Specifically, we reported 16 RNA-seq datasets for AlgR, CdpR, ExsA, GacA, LasR, MvfR, QscR, RhlR, RsaL, VqsM, VqsR, PhoB, GbdR, PchR, SphR and FleQ, as well as 12 ChIP-seq datasets for ExsA, GacA, MexT, MvfR, QscR, RhlR, SoxR, PhoB, GbdR, PchR, SphR and FleQ. These data are expected to provide valuable resources for this field of research. (3) We verified our RNA-seq and ChIP-seq results through EMSA, qRT-PCR and statistical evaluations. (4) Finally, we also developed a freely accessible website (http://pagnetwork.org/) and R package based on PAGnet. We hope that these tools will serve as a useful database and platform for the community.

To normalise the growth conditions ($OD_{600} = 0.6$ in LB) for all mutants, we performed 10 RNA-seq analyses (AlgR[58], ExsA[61], GacA[62], MvfR[45,71], RsaL[63], VqsM[72], VqsR[60], PhoB[73], GbdR[74] and SphR[75]) for which the regulons had been published previously in other strains and under other conditions. Compared with the previous results, the present study significantly extended the regulons of AlgR, ExsA, GacA, RsaL, PhoB, GbdR and SphR (see Supplementary Table 3 for a comparison of the results). For example, we found that AlgR had 201 DEGs, compared with 47 DEGs in the previous study[58]. We also found that AlgR regulated the activity of elastase (encoded by *lasB*), the *nirN-S* operon (involved in haeme biosynthesis), *nosR-L* operon (involved in denitrification pathway), *narL* and *narK1-I* (nitrogen metabolism). In the present study, we found that ExsA regulated 1,249 DEGs, a significantly greater number than the previous regulon[61]. Notably, ExsA regulated important non-T3SS pathways, including *Pqs-QS* (*pqsA-E/phnAB*), *pprB* (regulates biofilm formation and cell adhesion), *pchA-G/fptA* (pyochelin/siderophore pathway), the *spuA-H* operon (spermidine catabolic process), *aotJ-argR* (arginine biosynthetic and catabolic process) and *siaD/ptrR/N* (cyclic-di-GMP and biofilm). Strikingly, under our current growth conditions, PhoB regulated 1828 DEGs, including multiple virulence regulons such as PQS-QS (*mvfR/pqsA-phnAB*), Rhl-QS (*rhlAB* and *rhlRI*), T6SS (*tssL1-K1, tagQ1-icmF1, tagJ1-vgrG1, tse6, vgrG4b* and *vgrG6*) and glycine betaine catabolism (*gbdR*). A previous study elucidated the crosstalk between PhoB and TctD under phosphate-limited conditions[73].

The present study, which focused on virulence-related TFs, defined 347 direct target genes in a regulome of 20 TFs. We quantified the relative contributions of these TFs to the overall transcriptome plasticity of *P. aeruginosa* with the aim of revealing the architecture of the TF regulons and gaining a more comprehensive understanding of the virulence-related transcriptional network. Subsequently, we assigned the direct crosstalk preferentially to the biofilm formation system. RsaL, QscR, RhlR, CdpR, MvfR, LasR, PhoB and PchR were identified as master regulators of QS in our network (Hypergeometric test, BH-adjusted $P < 0.05$). We believe that our study has provided a valuable online platform that will allow the efficient integration of new data and facilitate future studies of the complex regulatory mechanisms associated with *P. aeruginosa* virulence and metabolism.

Although the known functions of these TFs were characterised previously, the present PAGnet identified some new functions. In addition to activating the induction of H1, H2 and H3-T6SS[62], we newly discovered that GacA regulated multiple pathways by directly binding to the promoters of *cdpR, pelB, glnD, nasA, mifS, napF, cbpD* and *magD*. PAGnet also revealed that 5 genes were directly co-regulated by GacA and other TFs. Moreover, we newly identified a group of genes (such as *popN*, PA3840, PA3842, *fabG, ccoN2* and *phrS*) that were directly regulated by ExsA. In this crosstalk, the AlgZR two-component AlgZR system also acted as a negative regulator of T3SS gene expression. Altogether, we revealed a complex relationship of interactions between these TFs and between the T3SS, T6SS and other virulence pathways.

Collectively, this study mapped an integrated regulatory network, PAGnet, comprising the interactive regulons of 20 TFs. This network provided new insights regarding functional crosstalk and revealed new potential functions of several TFs, including PhoB, FleQ, AlgR, ExsA and GacA. Furthermore, the present study yielded a valuable online platform and R package that will enable the efficient integration of new data and user-customised analyses online or offline. In summary, this study revealed a highly modular structure of complex crosstalk among virulence-related TFs and clearly depicted the complicated network regulating *P. aeruginosa* pathogenicity. The development of inhibitors against these newly identified master regulators may lead to the discovery of novel drugs targeting *P. aeruginosa*. The methodology and conclusions of this work may be applicable to other bacterial pathogens in the future.

## Methods

**Bacterial strains, culture media, plasmids and primers**. The bacterial strains, plasmids and primers are used in this study are listed in Supplementary Data 5. The *P. aeruginosa* PAO1 strain and its derivatives were grown at 37 °C in LB (Luria-Bertani) broth with shaking at 220 rpm or on LB agar plates. Antibiotics were used for *Escherichia coli* at the following concentrations: kanamycin at 50 μg/ml and ampicillin at 100 μg/ml.

**Deletion mutant construction**. A SacB-based strategy was employed for the construction of gene knockout mutants[19,76]. The pEX18AP plasmids was digested by using *Eco*RI and *Hind*III. The upstream(~1500-bp) and downstream (~1000-bp) of TF opening reading frame were amplified from PAO1 genome and digested with *Xba*I respectively (All primers are listed in Supplementary Data 5). Then the *Xba*I digested upstream(~1500-bp) and downstream fragments were ligated with T4 DNA ligase, the ligated DNA products were inserted into the *Eco*RI and *Hind*III digested pEX18AP plasmids using ClonExpress MultiS One Step Cloning Kit (Vazyme, China) generating the pEX18AP-TF plasmid. pEX18AP-TF was digested by *Xba*I and a 0.9 kb gentamicin resistance cassette cut from pPS858 with *Xba*I was then cloned into pEX18AP-TF, yielding pEX18AP-TF-Gm. The resultant plasmids were electroporated into PAO1 with selection for gentamicin resistance. Colonies were selected for gentamicin resistance and loss of sucrose (5 %) susceptibility on LB agar plates containing 50 μg/ml gentamicin and 5 % sucrose, which typically indicates a double-cross-over event and thus gene replacement. The Δ*rsaL* mutant was further confirmed by PCR.

**ChIP-seq analyses**. For the VSVG-tagged and FLAG-tagged plasmids, the ORF was amplified by PCR from PAO1 genome and cloned into pAK1900 plasmid by *Hind*III/*Bam*HI for the overexpressed TFs through *Hind*III site by using ClonExpress MultiS One Step Cloning Kit (Vazyme, China). Wild-type *P. aeruginosa* containing empty pAK1900 or pAK1900-TF-VSV-G/FLAG was cultured in LB medium supplemented with ampicillin until mid-log phase ($OD_{600} = 0.6$), then treated with 1 % formaldehyde for 10 min at 37 °C. Cross-linking was stopped by the addition of 125 mM glycine. Bacterial pellets were washed twice with a Tris buffer (20 mM Tris-HCl [pH 7.5] and 150 mM NaCl), re-suspended in 500 μl IP buffer (50 mM HEPES–KOH [pH 7.5], 150 mM NaCl, 1 mM EDTA, 1 % Triton X-100, 0.1% sodium deoxycholate, 0.1% SDS, and mini-protease inhibitor cocktail (Roche), and then subjected to sonication to produce 100-300 bp DNA fragments. Insoluble cellular debris was removed by centrifugation at 4 °C and the supernatant was used as the input sample in IP experiments. Both control and IP samples were washed with protein A beads (General Electric) and then incubated with 50 μl agarose- conjugated anti-VSV antibodies (Sigma) in IP buffer. Washing, crosslink reversal, and purification of the ChIP DNA were conducted[77]. DNA fragments (150-250 bp) were selected for library construction and sequencing libraries were prepared using the NEXTflex™ ChIP-Seq Kit (Bioo Scientific). The libraries were sequenced using the HiSeq 2000 system (Illumina). All the experiment has two repeats. The two repeats were merged together for the following analyses. ChIP-seq reads were mapped to the *P. aeruginosa* genomes (NC_002516) using Bowtie (Version 1.2.2). Only the uniquely mapped reads were kept for the subsequent analyses. Binding peaks ($P < 1 \times 10^{-5}$) were identified using MACS software (version 2.1.0). Consensus Motifs were identified using MEME with all significant peaks as input. TF target genes were annotated by peaks locating in gene promoters (upstream of gene start site or overlapping with gene start site).

A permutation test was performed to evaluate the statistical significance of the co-occurrence (co-binding) of multiple TFs on the same promoter of a gene using the following steps: (i) For each of the TFs co-binding a particular promoter of interest, we counted the real number of binding sites for the TF on all promoters in the entire genome and randomly distributed the same number of binding sites to all promoters. (ii) Step (i) was iterated $10^5$ times to generate an empirical background distribution of co-occurrence of the TFs on the promoter of interest. (iii) Based on the empirical background distribution, we calculated the number of co-occurrence (or co-binding)

events by chance (termed N) on the promoter of interest. iv) *P*-values were derived from $N/10^5$. (v) The raw *p*-values were finally adjusted for multiple hypothesis testing by the Benjamini–Hochberg procedure.

The ChIP-seq data files have been deposited into Gene Expression Omnibus (GEO) and can be accessed through GEO Series accession number GSE121243 and GSE128430.

**RNA-seq analyses**. To examine the effect of TFs (taking an example of RpoN) on the transcriptome, 2 ml of mid-log-phase ($OD_{600} = 0.6$) bacterial cultures (PAO1 and TF mutant strains) were collected by centrifugation (12,000 rpm, 4 °C). A RNeasy mini kit (Qiagen) was used for subsequent RNA purification with DNaseI (NEB) treatment. After removing rRNA by using the MICROBExpress kit (Ambion), mRNA was used to generate the cDNA library according to the NEBNext® UltraTM II RNA Library Prep Kit protocol (Illumina), which was then sequenced using the HiSeq 2000 system (Illumina). Each sample in RNA-seq assay was repeated twice. RNA-seq reads were mapped to the *P. aeruginosa* genomes (NC_002516) using STAR, only the uniquely mapped reads were kept for the subsequent analyses. Differentially expressed genes were identified using DESeq2 (BH-adjusted $P < 0.05$ and |log2 Fold Change| > 1)[78]. The statistical significance of co-occurrence of differential gene expression observed in multiple mutant TFs was evaluated by permutation tests similarly to what was performed in the ChIP-seq data analysis. All the experiment has two repeats. The RNA-seq datasets have been deposited in National Center for Biotechnology Information (NCBI) with an accession number GSE121243 and GSE128430.

**Quantitative RT-qPCR**. For real-time quantitative PCR (RT-qPCR), all strains were cultured at 37 °C, 220 rpm overnight in LB until $OD_{600}$ to 0.6. To harvest the bacteria, the cultures were centrifuged as pallets at 8000 rpm for 1 min. RNA purification was performed by using RNeasy minikit (Qiagen). RNA concentration was measured by Nanodrop 2000 spectrophotometer (ThermoFisher). The cDNA synthesis was performed by using a FastKing RT Kit (Tiangen Biotech). RT-qPCR was performed by SuperReal Premix Plus (SYBR Green) Kit (Tiangen Biotech) and prepared by following the manufacturer's instruction. Each reaction was performed in triplicate in 25 μl reaction volumes with 800 ng cDNA and 16S rRNA as an internal control. For each reaction, 200 nM primers (Supplementary Data 5) were used for RT-qPCR. The reactions were run at 42 °C for 15 min, 95 °C for 3 min, and kept at 4 °C until used. The fold change represents relative expression level of mRNA, which can be estimated by the values of $2^{-(\Delta\Delta Ct)}$. All the reactions were conducted with three repeats.

**Cloning and recombinant protein purification**. Oligonucleotides, restriction enzymes and vectors used for cloning of His tagged proteins analysed in this study are listed in Supplementary Data 5. cDNA clones acquired from the *P. aeruginosa* genome DNA amplified by polymerase chain reaction (PCR) to obtain the region encoding the protein (PhoB, FleQ, AmrZ, SoxR, GacA and ExsA). The PCR products were inserted into pMCSG19 vector[79] and transformed into *E. coli* BL21 (DE3) strain carrying pRK1037 plasmid[80]. The pET28a vector was transformed into *E. coli* BL21 (DE3) strain. Briefly, a single colony on the plate was inoculated into 10 ml sterilized LB broth containing 100 μg/ml ampicillin and 50 μg/ml kanamycin for 12 h. Then, we transferred the culture into 1 L the same medium as above and the cells were grown at 37 °C, 220 rpm to $OD_{600} = 0.6$. 0.5 mM IPTG (Isopropyl β-D-1-Thiogalactopyranoside) was added into the culture to induce protein expression at 16 °C for 16 h. The culture was centrifuged at 4 °C, 7000 rpm, for 5 min to harvest the bacteria. The whole processes were performed at 4 °C. The pellet was suspended in 20 ml buffer A (500 mM NaCl, 25 mM Tris-HCl, pH 7.4, 5% glycerol, 1 mM dithiothreitol, 1 mM PMSF (phenyl-methanesulfonyl fluoride)). The cells were lysed with sonication at six seconds interval and centrifuged at 4 °C (12000 rpm, 30 min). The supernatant was filtered with a 0.45 μm filter and the filtrate was added into a Ni-NTA column (Bio-Rad) which had been equilibrated with buffer A before using. After the Ni-NTA column was washed three times with buffer A, the column was eluted with 30 ml gradient of 60-500 mM imidazole prepared in buffer A respectively. Fractions from 300 mM to 500 mM were pooled and sodium dodecyl sulfate-polyacrylamide gel electrophoresis (SDS-PAGE) was used to verify the molecular weight of target protein.

**Electrophoretic mobility-shift assay**. DNA probes were PCR-amplified using primers listed in Supplementary Data 5. The probe (40 ng) was mixed with various amounts of protein in 20 μl of gel shift buffer (10-mM Tris-HCl, pH 7.4, 50-mM KCl, 5-mM MgCl₂, 10% glycerol). After incubation at room temperature for 20 min, the samples were analysed by 6% polyacrylamide gel electrophoresis (90 V for 60 min for sample separation). The gels were subjected to DNA dye for 5 min and photographed by using a gel imaging system (Bio-Rad). The assay was repeated at least three times with similar results.

**Mapping PAGnet**. Virulence regulatory network was mapped by integrating regulatory relationships experimentally validated in the literature or newly identified in our integrative analysis of ChIP-seq and RNA-seq data. For transcription factors with ChIP-seq and mutant RNA-seq data generated from our lab or known differentially expressed genes from previous studies, the transcriptional targets that are differentially expressed were identified as functional targets (Regulons).

Experimentally validated regulons (transcriptional factors and their targets) were integrated to virulence regulatory network directly. The statistical significance of a gene functionally co-regulated by multiple TFs was evaluated by permutation tests similarly to what was performed in the ChIP-seq data analysis. Master regulator analyses for QS, T3SS and T6SS were performed identified using Hypergeometric test (BH-adjusted $P < 0.05$). R package RedeR was used to visualize the network[81].

**Statistical analysis**. Two-tailed Student's $t$-tests were performed using Microsoft Office Excel 2010. $*P < 0.05$, $**P < 0.01$ and $***P < 0.001$ and results represent means ± SD. All experiments were repeated at least three times.

**Reporting summary**. Further information on research design is available in the Nature Research Reporting Summary linked to this article.

## Data availability

ChIP-seq and RNA-seq data are available in the National Center for Biotechnology Information Gene Expression Omnibus database under accession codes GSE121243 and GSE128430.

## Code availability

The code used for the entire data analysis process is freely available on github (https://github.com/CityUHK-CompBio/PAGnet_code).

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

## Acknowledgements

This study was supported by Health Medical Research Fund of Hong Kong (17160022 to Xin Deng), National Natural Science Foundation of China Grants (31670127 to Xin Deng and 81802384 to Xin Wang), General Research Funds of Hong Kong (21101115, 11102317, 11103718 to Xin Wang).

## Author contributions

X.D. and X.W. designed the project and obtained funding, material support and study supervision. X.D., X.W., HH., X.S. and Y.X. wrote the manuscript and generate the figures and tables. H.H. performed bioinformatic analysis. X.S. and Y.X. performed the experiments. T.W. and Y.Z. helped to edit the manuscript. All authors reviewed the manuscript.

## Additional information

**Competing interests:** The authors declare no competing interests.

