## [Peer Review File · Nature Communications]

Reviewers' comments:

Reviewer #1 (Remarks to the Author):

In this manuscript, Huang et al. present new ChIP-Seq and RNA-Seq results that they combine with previous data from their group and others in the field to generate what they term is the virulence network of *P. aeruginosa* (Pa). I think that this data will be useful to the field and this manuscript brings together and extends much useful data on virulence in Pa. There are a number of caveats, however, that diminish my enthusiasm for the manuscript:

1- Not all known virulence is captured: I realize that these authors, like all of us, are limited by time and/or resources, and cannot do everything. However, it was strange that regulators known to dramatically alter virulence were not included... I'm specifically thinking about PhoB, GbdR, AlgR, Fecl, PchR, SphR, FleQ, among many others. While the authors grabbed most of the major regulators, they missed too many to justify calling this a real integrated network of Pa virulence.

2- RpoN is not a transcription factor, it is a sigma factor. Since a sigma factor is a required component of RNA polymerase, it is not surprising that this particular sigma factor regulates the largest number of genes of all of the "TFs". This is the only sigma factor on the authors virulence list. They either need to remove it from the analysis and focus just on strict TFs, or, if the authors are going to include sigma factors into their analysis, they need to add FoxR, AlgU, FliA, and RpoH minimally, since all are important for virulence. This should be an easy fix, since the data used in this current manuscript to generate the network have already been generated by Schulz et al. 2015 (PLoS Pathogens), including ChIP-Seq, RNA-Seq, and binding site analysis for most of the non-essential sigma factors in Pa.

3- The authors provide no statistical analysis of their interaction networks. Since these regulators are known to have different scales of binding sites in the genome and there are limited upstream regions on which to bind, by simple network statistics, we'd predict overlapping binding at some frequency of promoters. The co-bound and co-regulated extinction curves (captured readily in the data from Supp Tables 2-4) seem to match a standard Gaussian extinction curve. Please provide statistical support that these overlaps are not simply by chance.

4- Transcriptomes vs Reality – One of the issues with the transcriptomes as analyzed in this manuscript is that most of the transcriptomes are derived from different experimental conditions, timepoints, and even strains of Pa. This really makes the analysis of co-regulated genes meaningless, since they might link more to shared responses of the mutants to experimental variables than direct regulation by those regulators.

5- Oddly, the authors have listed LipA, and LipC, known Type II secreted lipases, in their T6SS list... what is the rationale for this inclusion?

6- The methods section is not complete. The authors state they used anti-vsvG to do their ChIP, but do not describe the generation of the vsvG-tagged strains (or it is not detailed appropriately), and do not demonstrate that the vsvG-tagged versions complement the deletion strains. Without genetic complementation by the vsvG-tagged TFs, the ChIP-Seq is meaningless.

7- The grammar and syntax are nearly unreadable, mostly for the Introduction. Things get a little better in the Results, but this manuscript needs very heavy editing by a scientifically literate editor with English fluency.

Minor issues:

- The tap(sheet) for Supplemental Table 2 contains the right data but is labeled within the sheet as Table S7

- Many of the EMSA figs, while convincing, are of poor resolution and should be improved.

- RpoN association with promoters is known to be altered when it is in the complex as a portion of RNA polymerase, this potentially invalidates the EMSAs conducted with purified RpoN.

- The PaVIRnet site does not appear to be much more useful than the supplemental figures and tables are, at least for the data presented. The website is capable of receiving user input data but I have two problems – since this is not sourced on an academic server, is there a source for

sustained support for this site? Second, what assurances do users have that any network data they upload will not be used by the authors or others?

- Fig 1a should have a log2 or log10 scaled y-axis to make distinction amongst the TFs apparent.

Reviewer #2 (Remarks to the Author):

In this manuscript, the authors present the results of a very comprehensive analysis of *Pseudomonas aeruginosa* regulatory genes, including 19 regulators analyzed by ChIP-seq and transcriptomics methods. The ChIP-data are corroborated by EMSA analysis of selected targets showing the the results are largely valid. Alltogether, the data represent a strong contribution for the attempts to elucidate the highly complex regulatory networks of *P. aeruginosa*, a model organism that harbors around 500 regulatory genes (which is ~ 10% of the genomic content), which should also be of interest to a broader community.

However, despite the high value of this work, many key aspects of the presentation should be reconsidered and I will concentrate on these in this review.

1) I find both the term „PAVIRnet“ and the way it is introduced highly misleading. Firstly, it is not clear, why the network should be considered as a network regulating virulence. It is never clearly stated, why the authors selected the 19 regulators that were included here and the selection seems rather arbitrary. There is nothing wrong with that but this large number of regulators covers a major part of the genome which of course regulates virulence but also all kinds of other pathways. RpoN for example is mainly described as a regulator for nitrogen metabolism, which has many connections to virulence genes but not exclusively. I'd argue that it is even hard to find any regulatory gene affecting multiple operons that has not any connection to virulence at all.

Therefore, I think that the term „virulence“ is close to meaningless in the title and might as well be replaced by „genomic“. Secondly, speaking of „the PAVIRnet“ implies that it represents a functional unit (which the authors claim to have „constructed“), while in fact it is a rather arbitrary subset of the complete genomic regulatory network of *P. aeruginosa*. All this claims much more than the data can actually delivered and it should be tuned down accordingly.

2) Complex regulatory networks of *P. aeruginosa* have been published before, covering different aspects of the story and using different types of analyses (e.g., Schulz et al, 2015, doi: 10.1371/journal.ppat.1004744; Balasubramanian et al, 2013, doi: 10.1093/nar/gks1039). What are the new aspects covered by this work that have been missing before? It is not clearly stated and the discussion should better put this work in the context of previous knowledge.

3) The writing appears somewhat sloppy in parts of the manuscript. The first sentences of the introduction are difficult to read due to mixed up wording or grammar. The nomenclature is very inconsistent, e.g., the use of the term „transcription factor“ (TF). Not all of the mentioned regulators are TFs, some regulate on post-transcriptional levels as the authors state themselves (e.g. line 104) and the list of regulators itself seems to vary: line 95 mentions WspR, VqsR and BfmR that are not mentioned at all in the next paragraph. Instead, in line 109 ExsA appears, which was not mentioned before. The text should be carefully proofread and checked for such inconsistencies and also to remove the many smaller typos and language errors.

4) There are also inconsistencies between the descriptions of results. i) The network was built on 19 „TFs“ (line 177) by assigning function targets based on two criteria, one of which was direct binding of the TF to the upstream region. This is in contradiction to the statement that at least three of the regulators act post-transcriptionally. Or another example ii) Table S4 and the text (line 185) claim that rhIR is co-regulated by multiple genes, however the ChIP-results only show binding of the rhIR promoter by RhIR itself. It appears as if binding was not really used as a hard criterion but the text does not reflect this.

5) There is no definition of „promoter region“ or „upstream of genes“ and „downstream of genes“, which all is absolutely necessary to understand, how the ChIP-results were associated with gene loci.

- 6) The section „Experimental verification of PAVIRnet“ contains misleading statements. Firstly, it is not an experimental verification of the network as such but of a subset of the results from ChIP-seq and transcriptome analysis. Secondly, no „predicted regulatory relationships“ are presented (line 230). The network is deduced from the data and does not predict any additional relationships.
- 7) The discussion is mainly repeating many results instead of putting the data into context.
- 8) The supplementary tables are not sufficiently explained, which limits their use

We want to thank the reviewers for their positive, insightful and constructive comments. During the past four months, we have done a series of experiments (16 RNA-seq, 5 ChIP-seq, biochemical verifications and statistical analyses) and carefully made substantial revisions based on these comments. We updated all figures and tables. We have also significantly revised the main text per the reviewers' suggestions, which are highlighted in red in the revised manuscript. The following are our point-by-point responses to the reviewers' comments.

Reviewer #1 (Remarks to the Author):

In this manuscript, Huang et al. present new ChIP-Seq and RNA-Seq results that they combine with previous data from their group and others in the field to generate what they term is the virulence network of *P. aeruginosa* (Pa). I think that this data will be useful to the field and this manuscript brings together and extends much useful data on virulence in Pa. There are a number of caveats, however, that diminish my enthusiasm for the manuscript:

Response: Thank you for your positive and critical comments. We have extensively revised the following part based on your constructive comments.

1- Not all known virulence is captured: I realize that these authors, like all of us, are limited by time and/or resources, and cannot do everything. However, it was strange that regulators known to dramatically alter virulence were not included... I'm specifically thinking about PhoB, GbdR, FecI, PchR, SphR, FleQ, among many others. While the authors grabbed most of the major regulators, they missed too many to justify calling this a real integrated network of Pa virulence.

Response: We agree with the reviewer and apologise for missing many known important virulence factors from the literature. Thank you for your constructive suggestion on adding PhoB, GbdR, FecI, PchR, SphR and FleQ. We have performed ChIP-seq and RNA-seq for five new deletion strains, namely PhoB,

GbdR, PchR, SphR and FleQ. We have not added the sigma factor FecI (Schulz et al. PLoS Pathog. 2015) because we decided to exclude sigma factors in this network in accordance with the reviewer #1's second comment. Please check the following table for the summary of new data. We have incorporated the new data into the network and have revised all figures and tables. Please see Lines 126-131 and check the updated network in Figure 4.

No.	TFs	ChIP-seq	Condition	References
16	PhoB	PAO1	LB, OD = 0.6	This study
17	GbdR	PAO1	LB, OD = 0.6	This study
18	PchR	PAO1	LB, OD = 0.6	This study
19	SphR	PAO1	LB, OD = 0.6	This study
20	FleQ	PAO1	LB, OD = 0.6	This study

2- RpoN is not a transcription factor, it is a sigma factor. Since a sigma factor is a required component of RNA polymerase, it is not surprising that this particular sigma factor regulates the largest number of genes of all of the “TFs”. This is the only sigma factor on the authors’ virulence list. They either need to remove it from the analysis and focus just on strict TFs, or, if the authors are going to include sigma factors into their analysis, they need to add FoxR, AlgU, FliA, and RpoH minimally, since all are important for virulence. This should be an easy fix, since the data used in this current manuscript to generate the network have already been generated by Schulz et al. 2015 (PLoS Pathogens), including ChIP-Seq, RNA-Seq, and binding site analysis for most of the non-essential sigma factors in Pa.

Response: Thank you for the helpful comment. As the network of sigma factors has been reported previously (Schulz et al. PLoS Pathog. 2015), we decided to remove RpoN from the present study. In the revised manuscript, we have added five new transcription factors (TFs), namely PhoB, GbdR, PchR, SphR and FleQ, and have removed three non-TFs, namely QteE, RsmN and RsmA, thus resulting in 20 strict TFs. Please check the following table for the details. and we also listed the

information in Supplementary Table S2. All the ChIP-seq peaks and calculated binding motifs of TFs were shown in Figure 2. Please see Lines 168-179.

No.	TFs	ChIP-seq	Condition	References
1	AlgR	PAO1	LB, OD = 0.6	Our previous study (Kong et al. Nucleic Acids Res. 2015)
2	AmrZ	PAO1	LB, OD = 0.6	Jones et al. PLoS Pathog. 2014
3	BfmR	PAO1, ChIP-cloning	Biofilms	Petrova et al. Mol Microbiol. 2011
4	CdpR	PAO1	LB, OD = 0.6	Our previous study (Zhao et al. PLoS Biol. 2016)
5	ExsA	PAO1	LB, OD = 0.6	This study
6	GacA	PAO1	LB, OD = 0.6	This study
7	LasR	PAO1, ChIP in vitro	LB, OD = 0.6	Gilbert et al. Mol Microbiol. 2009
8	MexT	PAO1	LB, OD = 0.6	This study
9	MvfR	PAO1	LB, OD = 0.6	This study
10	QscR	PAO1	LB, OD = 0.6	This study
11	RhlR	PAO1	LB, OD = 0.6	This study
12	RsaL	PAO1	LB, OD = 0.6	Kang et al. Nucleic Acids Res. 2017
13	SoxR	PAO1	LB, OD = 0.6	This study
14	VqsM	PAO1	LB, OD = 0.6	Our previous study Liang et al. Nucleic Acids Res. 2014
15	VqsR	PAO1, via motif-searching in the genome	LB, OD = 0.6	Our previous study (Liang et al. J Bacteriol. 2012)
16	PhoB	PAO1	LB, OD = 0.6	This study
17	GbdR	PAO1	LB, OD = 0.6	This study
18	PchR	PAO1	LB, OD = 0.6	This study
19	SphR	PAO1	LB, OD = 0.6	This study
20	FleQ	PAO1	LB, OD = 0.6	This study

3- The authors provide no statistical analysis of their interaction networks. Since these regulators are known to have different scales of binding sites in the genome and there

are limited upstream regions on which to bind, by simple network statistics, we'd predict overlapping binding at some frequency of promoters. The co-bound and co-regulated extinction curves (captured readily in the data from Supp Tables 2-4) seem to match a standard Gaussian extinction curve. Please provide statistical support that these overlaps are not simply by chance.

Response: Thank you very much for the suggestion to perform statistical analysis for the network. Accordingly, we performed a permutation test to evaluate the statistical significance of the co-occurrence (co-binding) of multiple TFs on the same promoter of a gene using the following steps:

- 1. For each of the TFs co-binding a particular promoter of interest, we counted the real number of binding sites for the TF on all promoters in the entire genome and randomly distributed the same number of binding sites to all promoters.*
- 2. Step (1) was iterated 10^5 times to generate an empirical background distribution of co-occurrence of the TFs on the promoter of interest.*
- 3. Based on the empirical background distribution, we calculated the number of co-occurrence (or co-binding) events by chance (termed N) on the promoter of interest.*
- 4. P-values were derived from $N/10^5$.*
- 5. The raw p-values were finally adjusted for multiple hypothesis testing by the Benjamini–Hochberg procedure.*

Similar permutation tests were also performed to evaluate the statistical significance of co-occurrence of differential gene expression observed in multiple mutant TFs and a gene functionally co-regulated by multiple TFs. The raw p-values and adjusted p-values have been added to the corresponding Supplementary Tables 3–5 in the revised manuscript. Please see the Lines 486–496 and 511–514 in the Method section.

4- Transcriptomes vs Reality – One of the issues with the transcriptomes as analyzed in this manuscript is that most of the transcriptomes are derived from different experimental conditions, timepoints, and even strains of Pa. This really makes the

analysis of co-regulated genes meaningless, since they might link more to shared responses of the mutants to experimental variables than direct regulation by those regulators.

Response: We appreciate this critical comment, which is a key issue in this study. We agree that we need to standardise the experimental conditions used for all transcriptomes. Accordingly, we decided to use PAO1 as the background strain and OD₆₀₀ = 0.6 in the LB medium as the growth condition. We repeated the RNA-seq experiment for 16 deletion PAO1 strains (Δ phoB, Δ gbdR, Δ pchR, Δ sphR, Δ fleQ Δ cdpR, Δ gacA, Δ rsaL, Δ vqsR, Δ exsA, Δ lasR, Δ mvfR, Δ rhlR, Δ qscR, Δ vqsM and Δ algR) in the same condition (LB, OD₆₀₀ = 0.6). Please check the following table for detailed information and we also listed the information in Supplementary Table S2. Given that our previous study showed that BfmR regulates rhlQS only in the M8-glutamate minimal medium, but not in the LB medium (Cao et al. PLoS Pathog. 2014), we did not show its RNA-seq data in LB, OD₆₀₀ = 0.6.

All differentially expressed genes in all RNA-seq results and co-regulated genes among these TFs were shown in Figure 3, Figure S3 and supplementary table S4. Please see Lines 185-195.

No.	TFs	RNA-seq	Condition	References
1	AlgR	PAO1	LB, OD = 0.6	This study
2	AmrZ	PAO1	LB, OD = 0.6	Jones et al. PLoS Pathog. 2014
3	BfmR	PAO1, Microarray	Biofilms	Petrova et al. Mol Microbiol. 2011
4	CdpR	PAO1	LB, OD = 0.6	This study
5	ExsA	PAO1	LB+EGTA, OD = 0.6	This study
6	GacA	PAO1	LB, OD = 0.6	This study
7	LasR	PAO1	LB, OD = 0.6	This study
8	MexT	PAO1, Microarray	LB, OD = 0.6	Tian et al. Nucleic Acids Res. 2009
9	MvfR	PAO1	LB, OD = 0.6	This study
10	QscR	PAO1	LB, OD = 0.6	This study

11	RhlR	PAO1	LB, OD = 0.6	This study
12	RsaL	PAO1	LB, OD = 0.6	This study
13	SoxR	PAO1, Microarray	TSB+H2O2/PQ, OD = 0.6	Palma et al. Infect Immun. 2005
14	VqsM	PAO1	LB, OD = 0.6	This study
15	VqsR	PAO1	LB, OD = 0.6	This study
16	PhoB	PAO1	LB, OD = 0.6	This study
17	GbdR	PAO1	LB, OD = 0.6	This study
18	PchR	PAO1	LB, OD = 0.6	This study
19	SphR	PAO1	LB, OD = 0.6	This study
20	FleQ	PAO1	LB, OD = 0.6	This study

5- Oddly, the authors have listed LipA, and LipC, known Type II secreted lipases, in their T6SS list... what is the rationale for this inclusion?

Response: Response: We apologise for the errors. We have removed them in the revised manuscript.

6- The methods section is not complete. The authors state they used anti-vsvG to do their CHIP, but do not describe the generation of the vsvG-tagged strains (or it is not detailed appropriately), and do not demonstrate that the vsvG-tagged versions complement the deletion strains. Without genetic complementation by the vsvG-tagged TFs, the CHIP-Seq is meaningless.

Response: Thank you for bringing out this important point. In the Methods section of the revised manuscript, we have added a detailed description of the generation of vsvG-tagged strains. Please see Lines 460-463 in the Method section.

We performed a pyocyanin production assay and observed that vsvG-tagged RsaL and MyfR functionally complement their corresponding deletion strains (Kang et al. Nucleic Acids Res. 2017; Zaborina et al. PLoS Pathog. 2007). A Congo red assay also showed that FLAG-tagged FleQ indeed complemented the colony morphology of Δ fleQ strain (Hickman et al. Mol Microbiol. 2008). The results are displayed in Figure S1. Please see Lines 131-134.

7- The grammar and syntax are nearly unreadable, mostly for the Introduction. Things get a little better in the Results, but this manuscript needs very heavy editing by a scientifically literate editor with English fluency.

Response: We apologise for the language errors. The revised manuscript has been significantly edited by a scientific editor who is native speaker of English. The Introduction section has been largely re-written. Please see Lines 44-112.

Minor issues:

- The tap(sheet) for Supplemental Table S2 contains the right data but is labeled within the sheet as Table S7.

Response: Thank you for pointing out the error. We have corrected the label for Table S3 in the revised manuscript. Please see Lines 152, 166 and 609.

- Many of the EMSA figs, while convincing, are of poor resolution and should be improved.

Response: We agree with your comment. Accordingly, we have replaced the EMSA figures with higher resolution images. The EMSA results were redone and shown in Figures 6, S7 and S8. Please see Lines 21-276.

- RpoN association with promoters is known to be altered when it is in the complex as a portion of RNA polymerase, this potentially invalidates the EMSAs conducted with purified RpoN.

Response: We agree with your comment. In the revised manuscript, we have removed all results on RpoN in accordance with the reviewer #1's major comment #2.

- The PaVIRnet site does not appear to be much more useful than the supplemental figures and tables are, at least for the data presented. The website is capable of receiving user input data but I have two problems – since this is not sourced on an academic server, is there a source for sustained support for this site? Second, what assurances do users have that any network data they upload will not be used by the authors or others?

Response: Thank you for the insightful comments. We redeployed the website onto an academic server of the City University of Hong Kong (<http://pagnet.org/>). The website will be maintained and updated regularly by us. We have declared on our website that we do not store or share any user-uploaded data and that we will protect the confidentiality and ensure compliance with academic ethics.

To provide more flexibility, we also made PAGnet available as an R package 'PAGnet' in Github that can be freely downloaded, locally installed and used by the user on his/her personal computer. Thus, the user can choose to use the R package locally should they have further concerns about the confidentiality of their data. All functions available on our website are also provided in the 'PAGnet' R package by running a local shiny GUI within R. In addition, the master regulator analysis can be performed using the function 'pagnet.mra' in the R console without running the shiny GUI. Detailed instructions about installation and step-by-step use of the package are provided in the vignette of the 'PAGnet' package. Please see Lines 347-355.

- Fig 1a should have a log2 or log10 scaled y-axis to make distinction amongst the TFs apparent.

Response: We agree with the comment. Accordingly, we have revised the y-axis showing log2 in Figure 1a. Please see the Figure 1a and Line 636.

Reviewer #2 (Remarks to the Author):

In this manuscript, the authors present the results of a very comprehensive analysis of *Pseudomonas aeruginosa* regulatory genes, including 19 regulators analyzed by ChIP-seq and transcriptomics methods. The ChIP-data are corroborated by EMSA analysis of selected targets showing the results are largely valid. Altogether, the data represent a strong contribution for the attempts to elucidate the highly complex regulatory networks of *P. aeruginosa*, a model organism that harbors around 500 regulatory genes (which is ~ 10% of the genomic content), which should also be of interest to a broader community.

Response: Thank you for your positive comments!

However, despite the high value of this work, many key aspects of the presentation should be reconsidered and I will concentrate on these in this review.

1) I find both the term “PAVIRnet” and the way it is introduced highly misleading. Firstly, it is not clear, why the network should be considered as a network regulating virulence. It is never clearly stated, why the authors selected the 19 regulators that were included here and the selection seems rather arbitrary. There is nothing wrong with that but this large number of regulators covers a major part of the genome which of course regulates virulence but also all kinds of other pathways. RpoN for example is mainly described as a regulator for nitrogen metabolism, which has many connections to virulence genes but not exclusively. I’d argue that it is even hard to find any regulatory gene affecting multiple operons that has not any connection to virulence at all. Therefore, I think that the term “virulence” is close to meaningless in the title and might as well be replaced by “genomic”.

Secondly, speaking of “the PAVIRnet” implies that it represents a functional unit (which the authors claim to have “constructed”), while in fact it is a rather arbitrary subset of the complete genomic regulatory network of *P. aeruginosa*. All this claims much more than the data can actually delivered and it should be tuned down accordingly.

Response: We appreciate the critical and constructive comments and agree with them. In pathogenic bacteria, virulence is highly correlated with metabolism, so it is misleading to name the network 'virulence'. Throughout the revised manuscript, we have revised the name of the network to 'PAGnet', in which 'virulence' is replaced with 'genomic'. Please see Lines 1, 32, 103, 105, 235, 255 and throughout the manuscript. Based on reviewer #1's comment on sigma factor, we have removed RpoN from the network.

We agree that this study deals with only an arbitrary group of the complete genomic regulatory network of P. aeruginosa. The word 'constructed' is not appropriate. We have removed it from the revised manuscript and have revised the related statements. We think that this study presents a small database or platform that we can continue to add new results into and will hopefully contribute to the future study on complicated regulatory mechanisms in P. aeruginosa virulence and metabolism. We have discussed this point in the Discussion section. Please see Lines 373-383, 411-414 and 425-435.

2) Complex regulatory networks of P. aeruginosa have been published before, covering different aspects of the story and using different types of analyses (e.g., Schulz et al, 2015, doi: 10.1371/journal.ppat.1004744; Balasubramanian et al, 2013, doi: 10.1093/nar/gks1039). What are the new aspects covered by this work that have been missing before? It is not clearly stated and the discussion should better put this work in the context of previous knowledge.

Response: Thank you for the important comment. We apologise for not highlighting our new findings and comparison between this study and previous two high-quality studies. Schulz et al. focused on the regulatory network of 10 sigma factors (AlgU, FliA, RpoH, RpoN, RpoS, PvdS, FpvI, FecI, SigX and FecI2) in response to challenging conditions, which showed that the modular architecture of sigma factor networks enables adequate P. aeruginosa function in its environment. In 2013, Balasubramanian et al. comprehensively summarised the previous studies

on virulence factors and provided a global picture of virulence gene regulation in P. aeruginosa.

Compared with these two papers, our study presented new information in the following aspects: 1) Although the study by Schulz et al. focused on sigma factors in PA14, the present study focused strictly on TFs in PAO1. 2) Most importantly, we reported a large set of new data which was not included in the previous 2 studies. Specifically, we reported 16 RNA-seq datasets for AlgR, CdpR, ExsA, GacA, LasR, MvfR, QscR, RhlR, RsaL, VqsM, VqsR, PhoB, GbdR, PchR, SphR and FleQ, as well as 12 ChIP-seq datasets for ExsA, GacA, MexT, MvfR, QscR, RhlR, SoxR, PhoB, GbdR, PchR, SphR and FleQ. These data are expected to provide valuable resources for this field of research. 3) We verified our RNA-seq and ChIP-seq results through EMSA, qRT-PCR and statistical evaluations. 4) Finally, we also developed a freely accessible website (<http://pagnetwork.org/>) and R package based on PAGnet. We hope that these tools will serve as a useful database and platform for the community. We have discussed this point in the Discussion section. Please see Lines 90-94 and 368-383.

3) The writing appears somewhat sloppy in parts of the manuscript. The first sentences of the introduction are difficult to read due to mixed up wording or grammar. The nomenclature is very inconsistent, e.g., the use of the term “transcription factor” (TF). Not all of the mentioned regulators are TFs, some regulate on post-transcriptional levels as the authors state themselves (e.g. line 104) and the list of regulators itself seems to vary: line 95 mentions WspR, VqsR and BfmR that are not mentioned at all in the next paragraph. Instead, in line 109 ExsA appears, which was not mentioned before. The text should be carefully proofread and checked for such inconsistencies and also to remove the many smaller typos and language errors.

Response: We agree your comment and apologise for these language errors. The revised manuscript has been significantly edited by a scientific editor who is a native speaker of English. The first part of Introduction has been re-written. Please see Lines 44-50. In the revised manuscript, we have standardised the nomenclature

on ‘transcription factor’, which is the focus of the study. Based on the comments from both reviewers, we have removed non-TFs such as WspR, QteE, RsmN and RsmA. In addition, we have added introduction on VqsR, BfmR and ExsA in the Introduction section and Supplementary Table S1. Please see Lines 66, 73 and 95.

4) There are also inconsistencies between the descriptions of results. i) The network was built on 19 “TFs” (line 177) by assigning function targets based on two criteria, one of which was direct binding of the TF to the upstream region. This is in contradiction to the statement that at least three of the regulators act post-transcriptionally. Or another example ii) Table S4 and the text (line 185) claim that rhIR is co-regulated by multiple genes, however the ChIP-results only show binding of the rhIR promoter by RhIR itself. It appears as if binding was not really used as a hard criterion but the text does not reflect this.

Response: We agree with your comment and apologise for the inconsistencies in the descriptions of results. 1) In the revised manuscript, we have removed non-TFs such as QteE, RsmN and RsmA. Please see the Supplementary Table S1 for detailed information. 2) We apologise for the confusion on the co-regulation of rhIR by multiple TFs. ChIP-seq of RhIR showed only one target on its own promoter. Other TFs (such as BfmR, LasR and MvfR) could also bind to the same promoter of rhIR. We have clarified this in the revised manuscript. Please see Line 162 and Supplementary Table S3.

5) There is no definition of “promoter region” or “upstream of genes” and “downstream of genes”, which all is absolutely necessary to understand, how the ChIP-results were associated with gene loci.

Response: Thank you for the suggestion to clarify the definitions of genomic loci. We have provided clarifications in the revised manuscript. To illustrate these definitions, please see the following schematic diagram and Lines 140-144.

- *‘upstream of a gene’*: defined as the intergenic region upstream of a gene;
- *‘downstream of a gene’*: defined as the intergenic region downstream of a gene;
- *‘promoter region’*: also defined as the intergenic region upstream of a gene. This definition is based on previous studies (Liang et al. *Nucleic Acids Res.* 2014; Kong et al. *Nucleic Acids Res.* 2015; Zhao et al. *PLoS Biol.* 2016).

6) The section “Experimental verification of PAVIRnet” contains misleading statements. Firstly, it is not an experimental verification of the network as such but of a subset of the results from ChIP-seq and transcriptome analysis. Secondly, no “predicted regulatory relationships” are presented (line 230). The network is deduced from the data and does not predict any additional relationships.

Response: We agree with your comment and apologise for the misleading statements. In the revised manuscript, we have revised the expression ‘Experimental verification of PAGnet’ to ‘Experimental verification of the functional targets of TFs’. Please see Line 264. We have also revised the expression ‘predicted regulatory relationships’ to ‘validation’ and have removed the word ‘predicted’ throughout the manuscript. Please see Line 277.

7) The discussion is mainly repeating many results instead of putting the data into context.

Response: Thank you for the comment. We have re-written much of the Discussion section by adding our new data into the context. We have also discussed the differences between our study and previous studies and have highlighted our new finding. Please see Lines 368-403.

8) The supplementary tables are not sufficiently explained, which limits their use.

Response: We have added the following descriptions for the revised supplementary tables. And we added the descriptions of supplementary tables in the revised manuscript. Please see Lines 603-624.

*Table S1. Major functions of 20 transcription factors on *P. aeruginosa* virulence. The known functions of 20 regulators were summarised based on our study and other previous studies.*

Table S2. Experimental conditions for each of the ChIP-seq and RNA-seq transcription factors used in this study. The major biological pathway for each transcription factor is shown. Our samples were normalised at the mid-log phase ($OD_{600} = 0.6$) in the LB medium. Data from other studies are cited accordingly.

Table S3. Co-occurrence of multiple TFs revealed by ChIP-seq. Genes were co-bound by more than two transcription factors are shown with the statistical significance of the co-occurrence (co-binding) of multiple TFs on the same promoter of the gene.

Table S4. Co-occurrence of differentially expressed genes observed in multiple mutant TFs revealed by RNA-seq. Genes were co-regulated by more than two transcription factors are shown with the statistical significance of co-occurrence of differential gene expression observed in multiple mutant TFs.

Table S5. Genes functionally co-regulated by multiple TFs (PAGnet) based on ChIP-seq and RNA-seq. Genes were functionally co-regulated (intersection of Tables S3 and S4) by more than two transcription factors are shown with the statistical significance.

Table S6. Function enrichment of regulons in PAGnet. Hypergeometric tests ($P < 0.05$) were performed for each regulon, which is based on functional gene sets from PseudoCAP including Gene Ontology and KEGG databases.

Table S7. Bacterial strains, plasmids, and primers used in this study.

Table S8. Comparison between the RNA-seq data in this study and the previous studies.

REVIEWERS' COMMENTS:

Reviewer #1 (Remarks to the Author):

The authors have done a commendable job in generating substantial new and appropriate data in responding to my previous issues. The authors were overall responsive and the manuscript much improved.